# OpenReview forum: "Expectation Alignment of Language Models for Real-World User Expectations"
_ICML.cc/2026/Conference — ICML 2026 regular_

### Official Review · Reviewer_CaJf · 2026-03-12

**Soundness:** 3
**Presentation:** 3
**Significance:** 3
**Originality:** 3
**Overall Recommendation:** 4
**Confidence:** 3

**Summary:**

While large language models demonstrate strong performance on standard benchmarks, these benchmarks often rely on heuristics, expert-designed criteria, or simulated users, making it difficult to capture the diversity of real human expectations. To address this limitation, the authors analyze real multi-turn human–LLM conversations and propose a principled procedure for extracting semantically rich user expectations. Based on this procedure, they introduce EXPECTBENCH, a benchmark designed to evaluate whether models can satisfy and anticipate users’ desired content. Their analysis suggests that current large language models struggle to meet and predict user expectations. To mitigate this issue, the authors further propose LENS, a lightweight latent-expectation-aware response generation framework that aims to help models internalize user expectations and produce more aligned and consistent response.

**Compliance With Llm Reviewing Policy:**

Affirmed.

**Key Questions For Authors:**

**Sustainability and Reproducibility of the Evaluator Model:** The paper relies heavily on GPT-4o as an LLM-as-a-judge for automated evaluation across the benchmark. Considering that GPT-4o has been deprecated and removed by OpenAI, this poses a significant challenge to the reproducibility of the benchmark's baselines and the viability of future evaluations. Do the authors have plans to introduce or evaluate alternative models (e.g., newer closed-source APIs or high-performance open-source models) to re-validate or sustain the current evaluation pipeline?

**Limitations:**

Yes

**Strengths And Weaknesses:**

### Strengths

- **Highly Realistic Dataset with Rich Dimensions:** The EXPECTBENCH dataset constructed in this paper demonstrates high authenticity and complexity, fully considering multilingual coverage (encompassing 9 major languages), multi-turn dialogue depth (ranging from 3 to 15 turns), and multiple expectations within a single interaction (averaging 2.91 distinct expectations per query). This design successfully bridges the gap in existing benchmarks and accurately reflects the complex and multi-dimensional user intents and needs in real-world scenarios.
- **Clear and Rigorous Methodology for Dataset Construction:** The paper exhibits exceptional clarity and logical rigor in its dataset construction and evaluation methodology:
  - **Data Sourcing and Processing:** The research builds the dataset upon real, open-source human-AI interaction logs (WildChat) and implements a logically comprehensive data processing pipeline. This pipeline includes multi-turn filtering, query length control, task-level filtering, and human verification, effectively filtering out noise and ensuring high data quality.
  - **Evaluation Framework Construction:** The authors not only iteratively generated 10 rigorous expectation scoring dimensions through a data-driven approach , but also provided detailed experimental validation showing high consistency between the GPT-4o based automated evaluation and human expert annotations (achieving a Pearson correlation of 0.963 and a Cohen's Kappa of 0.626). This provides a solid backing for the benchmark's reliability.
- **Substantial Workload and Solid Contributions:** The workload presented in this study is highly sufficient. The authors not only constructed a large-scale benchmark comprising 12,000 instances and 34,876 extracted expectations , but also conducted a comprehensive evaluation of 6 representative mainstream LLMs. Furthermore, they proposed and validated the LENS framework to effectively address and mitigate the expectation alignment problem.

## Weakness

- **Potential Misalignment Between Experimental Setup and the Dataset's Core Premise:** The primary strength of EXPECTBENCH lies in capturing implicit or progressive user expectations through multi-turn interactions. However, in the Expectation Prediction experiment (Section 5.1) , the authors task the model with predicting expectations solely based on the initial query. This setup may introduce evaluation bias: on one hand, if a user introduces completely new requirements in follow-up turns that were never implied in the first turn, the model inherently cannot predict them, leading to an unfair underestimation of the Coverage metric. On the other hand, if a user exhaustively states all expectations in their initial query, the prediction task becomes trivial, potentially overestimating the metric. The dynamic evolution of expectations across turns is not adequately isolated in this evaluation, which may compromise the confidence and reliability of these specific results.

---

> ### Author Rebuttal · Authors · 2026-03-31
>
> We sincerely thank the reviewer for the thoughtful feedback and positive evaluation of our work.
>
> **1. Expectation Prediction Experimental Setup**
>
> We sincerely thank the reviewer for this thoughtful and valuable discussion. This point is important, and we would like to further clarify the role of Section 5.1 in our paper. Our intention in this section is not to use expectation prediction as the primary evaluation of the dataset itself, but rather as a targeted diagnostic analysis for two reasons. First, it helps us better understand the failure mode behind the results in Section 4: when models do not satisfy real user expectations, is the issue that they fail to infer the relevant criteria altogether, or that they roughly know the direction but still do not realize it well in the response? Second, it reflects the practical setting we care about: at response time, the model only has access to the initial context, and the question is whether it can anticipate the evaluative criteria that the user will later apply.
>
> For the first concern raised by the reviewer, namely completely new requirements introduced only in follow-up turns, such cases are already controlled during benchmark construction. We use explicit extraction constraints to distinguish expectations from new requests, together with manual review of instances. In addition, we add an extraction-level human validation on a random sample of 100 user queries (289 expectations) and find no cases of New / Not Anchored, providing direct evidence that the extracted expectations are not driven by newly introduced requests. This suggests that genuinely new requirements are unlikely to be a major source of distortion for the Section 5.1 analysis.
>
> For the second concern, namely cases where users may state their expectations very fully in the initial query, we believe this is not the dominant pattern in realistic interaction. EXPECTBENCH is built from real-world conversations precisely because user-valued criteria are often grounded in the initial context without being exhaustively enumerated. Even when the initial context is already rich and informative, the task is still not merely to restate explicit instructions, but to infer the evaluative criteria that the user will later apply. This is also consistent with the empirical results: the average initial query is already long (366 tokens), and each interaction contains only 2.91 expectations on average, yet we allow models to predict up to 10 expectations. Even under this permissive setting, the strongest model still achieves only around 40% coverage.
>
> Taken together, this indicates that the challenge in Section 5.1 mainly comes from anticipating context-grounded, implicit user criteria from the initial query alone. We hope this clarifies that Section 5.1 is intended to probe anticipatory expectation inference from the initial context, rather than to approximate the full process of multi-turn adaptation.
>
> **2. Alternative Evaluator**
>
> We sincerely thank the reviewer for raising this point. We fully agree that evaluator sustainability and reproducibility are important concerns. We would like to clarify that, while GPT-4o has been retired from ChatGPT, it remains available in the API, so the current evaluation pipeline is still reproducible as implemented.
>
> To further reduce dependence on a single evaluator, we additionally test Llama-3.3-70B-Instruct as a strong open-weight alternative judge. The resulting agreement remains high: $\kappa = 0.588 / r = 0.94$ against GPT-4o, and $\kappa = 0.543 / r = 0.96$ against human judgments. This suggests that our conclusions do not depend on one proprietary evaluator.
>
> More fundamentally, EXPECTBENCH is evaluator-agnostic by design: the benchmark is grounded in explicit user-derived expectation rubrics, and the judge’s role is only to score responses against these criteria rather than define them. This makes the evaluation pipeline straightforward to sustain with newer closed-source APIs or strong open-weight models, while preserving the same underlying evaluation target.

---

### Official Review · Reviewer_U4pT · 2026-03-13

**Soundness:** 2
**Presentation:** 3
**Significance:** 3
**Originality:** 3
**Overall Recommendation:** 4
**Confidence:** 4

**Summary:**

This paper studies the gap between LLM benchmark performance and real-world user satisfaction by extracting implicit user expectations from follow-up messages in multi-turn conversations. The authors introduce EXPECTBENCH and show that even GPT-4o scores only 2.72/5 on expectation alignment. They also propose LENS, a lightweight framework that injects latent expectation representations into a frozen LLM via a trained projector, yielding modest improvements.

**Compliance With Llm Reviewing Policy:**

Affirmed.

**Final Justification:**

The extraction validation study resolves the two most fundamental concerns. The benchmark contribution and expectation alignment formalization are quite convincing. LENS gains are modest but the primary contribution is the benchmark itself.

**Key Questions For Authors:**

1. What fraction of extracted expectations would human annotators confirm as genuinely implied (vs. new requests)? This directly determines benchmark validity.
2. How was extraction quality validated beyond downstream scoring agreement?
3. Does LENS help stronger base models (e.g. 70B), or do gains vanish at scale?
4. How stable is the dimension taxonomy across different random partitions and LLMs?

**Limitations:**

Partially addressed. The paper omits discussion of extraction reliability, cultural biases in WildChat, and the exclusion of coding/math tasks.

**Strengths And Weaknesses:**

## Strengths

- It is important and timely problem formulation. The insight that task completion does not equate to user satisfaction is well-argued, and formalizing expectations from follow-ups as evaluation criteria is creative and practically grounded.
- Well-designed benchmark. Multilingual coverage, realistic query lengths, multi-turn structure, and data-driven dimension discovery represent clear improvements over WildBench-style single-turn evaluations.
- Useful diagnostic decomposition. Separating expectation prediction from expectation satisfaction yields the striking finding that models cover only around 40% of expectations even at k=10, showing a root cause of misalignment.

## Weaknesses
- No validation of expectation extraction quality. The entire benchmark depends on GPT-4o correctly extracting expectations from follow-ups, yet there is no human evaluation of this critical step, only of downstream scoring.
- Conceptual conflation of unmet expectations with emergent preferences. Many follow-ups (e.g. "add that I returned from China") appear to introduce new ideas rather than reveal pre-existing expectations. This undermines the claim that models are fundamentally "misaligned."
- LENS gains are marginal and lack human validation. Improvements are small for Llama and barely exceed SFT-Resp (2.33). Evaluating LENS exclusively via GPT-4o, the very paradigm the paper critiques, is contradictory.

---

> ### Author Rebuttal · Authors · 2026-03-31
>
> We sincerely thank the reviewer for the thoughtful and constructive feedback.
>
> **1. Extraction Quality Validation**
>
> We agree that extraction quality is critical, since extracted expectations are the primary reference for evaluation. The current submission already includes multiple safeguards during benchmark construction: (1) data filtering to retain instances suitable for expectation-driven evaluation, (2) explicit extraction constraints that distinguish expectations from restatements, instructions, unsupported inferences, and new requests, and (3) manual review of every instance to verify overall quality and remove noisy cases.
>
> To address this concern more directly, we add an extraction-level human validation study on a random sample of 100 user queries (289 expectations). Three annotators independently label each expectation, with majority voting. We evaluate whether each expectation is supported by the follow-up and anchored to the initial request.
>
> We find that 95.5% are clearly supported by the follow-up, 4.5% are weakly but still plausibly supported, and 0% are unsupported. We further find that the extracted expectations remain strongly grounded in the initial request: 4.5% are Explicit Requirements (directly stated), 73.7% are Implicit Expectations (reasonably inferable), 16.6% are Refinement/Preference (further specification), 5.2% are Context-Utilization Needs (requiring better use of existing context), and 0% are New/Not Anchored. These results directly support the validity of our extraction step.
>
> **2. Unmet Expectations**
>
> We respectfully clarify that our benchmark does not treat arbitrary follow-up content as unmet expectations. The key question is whether a follow-up reveals a criterion already licensed by the original request and context, rather than a genuinely new goal. In the example, “add that I returned from China” is better understood as a Context-Utilization Need: the information is already present in the original materials, including the applicant’s location and the profile statement, “After 9 years in China, I am committed to bringing my versatile skillset to a high-energy US company promoting a positive community culture.” Emphasizing it is therefore not a new request, but a context-grounded detail that makes the response more useful for the user’s real goal.
>
> At the same time, genuinely new requests are explicitly controlled during benchmark construction through extraction constraints and manual review. As reported in Point 1, our added human validation further finds none labeled as New/Not Anchored, supporting that the extracted expectations are not driven by new requests.
>
> **3. Interpreting the LENS Gains**
>
> The comparison to SFT-Resp is in fact conservative for LENS. In SFT-Resp, GPT-4o first generates target responses with access to the full expectation set, and the entire response model is then fine-tuned on these expectation-aware outputs. By contrast, LENS does not fine-tune the whole response model; it only trains a lightweight projector to inject latent signals. Under this much weaker supervision and lower adaptation cost, LENS consistently exceeds SFT-Resp. Relative to base models, LENS improves by +9.2% on LLaMA and +4.6% on Mistral, with gains across all dimensions. This shows that LENS is a lightweight and effective way to improve expectation alignment.
>
> **4. GPT-4o as Evaluator**
>
> We respectfully clarify that this is not contradictory. Once the evaluation criteria are made explicit, the evaluator is no longer required to infer the user’s expectations itself, but only to judge responses against the given rubrics. This is a standard use of LLM-as-a-judge in recent work. We also quantify this in our setting: GPT-4o achieves substantial agreement with human judgments, with $\kappa=0.626$ and $r=0.963$ ($p<0.001$), supporting its use as a trustworthy evaluator in our benchmark.
>
> **5. LENS on Stronger Base Model**
>
> We additionally evaluate it with Llama-3.3-70B-Instruct as the main response model and LLaMA-3.1-8B-Instruct as the auxiliary observer. The overall score improves from 2.58 to 2.69, with gains across all ten dimensions: Cx: 2.59→2.71, Cr: 2.45→2.57, Co: 3.06→3.20, Cs: 2.59→2.72, Ct: 2.80→2.95, Ep: 2.82→2.95, Vs: 2.25→2.35, Pr: 2.41→2.52, En: 2.62→2.73, Cm: 2.42→2.53. This suggests that the benefit of LENS does not vanish at larger scale; the latent expectation signal remains useful for stronger models.
>
> **6. Taxonomy Stability**
>
> To assess taxonomy stability, we rerun the same iterative discovery procedure under three random partition orders and also with Llama-3.3-70B-Instruct. The overall taxonomy structure remains stable, with average similarity of 0.743 across random partitions and 0.672 across models. The remaining variation mainly comes from naming, phrasing, and granularity differences rather than changes in the underlying evaluative structure, e.g., “Personalization and Contextual Relevance” versus “Personalization and User Contextualization.”

---

> > ### Author Rebuttal · Reviewer_U4pT · 2026-04-04
> >
> > I appreciate the author's reply. Most of my concerns are resolved. I have raised my score.

---

> > > ### Author Response · Authors · 2026-04-06
> > >
> > > We sincerely thank the reviewer for finding our response helpful and for the valuable feedback on our work. We are especially glad that the clarification and additional analyses were useful, and we will make sure these points are reflected more clearly in the revised paper.

---

### Official Review · Reviewer_5asc · 2026-03-13

**Soundness:** 2
**Presentation:** 3
**Significance:** 3
**Originality:** 3
**Overall Recommendation:** 3
**Confidence:** 3

**Summary:**

This paper addresses the phenomenon where user needs are not fully articulated in single-turn interactions. Based on dialogue data between real human users and large language models (LLMs), the authors construct a benchmark that uses the fulfillment of user expectations as an evaluation metric. Six LLMs are assessed and analyzed, revealing shortcomings in both the understanding and alignment of user expectations by current LLMs. The paper further designs and implements a latent expectation-aware response generation framework, which improves the degree of user expectation fulfillment on the proposed benchmark.

**Compliance With Llm Reviewing Policy:**

Affirmed.

**Key Questions For Authors:**

See the above.

**Limitations:**

yes

**Strengths And Weaknesses:**

##Strengths
-1.The focus on incompletely expressed user expectations identifies a meaningful and underexplored research problem.
-2.The approach of reconstructing user expectations from real-world conversation data offers a practical methodology.

##Weaknesses
-1.User expectations may also be reflected in subsequent responses, particularly when certain expectations have already been fulfilled and are therefore not explicitly mentioned by the user. Constructing expectations solely based on user utterances may not fully capture this aspect.
-2.The annotated expectations serve as the primary reference for evaluation, and their accuracy should ideally be supported by manual verification—an aspect not addressed in the paper.
-3.Different users may hold varying expectations in response to the same query, and such user-dependent variation is not observable to the LLM. In this context, the predicted expectations involve a degree of stochasticity, making the metrics used in Section 5.1 potentially less suitable for evaluation. It might be more appropriate to assess expectation prediction at the level of expectation categories.
-4.When describing the latent expectation–aware response generation framework, it is not entirely clear how the Expectation Observer and the Expectation Projector are implemented—for instance, details regarding their specific model architectures and training procedures would help to clarify the approach.

---

> ### Author Rebuttal · Authors · 2026-03-31
>
> We sincerely thank the reviewer for the thoughtful and constructive feedback.
>
> **1. Follow-up-Revealed Expectations**
>
> We thank the reviewer for raising this important point. We agree that some user expectations may not be explicitly mentioned in later turns precisely because they have already been satisfied. However, such expectations are difficult to identify and verify reliably at scale without additional confirmation from the original user. EXPECTBENCH is therefore not designed to exhaustively enumerate every latent expectation, but to focus on those evaluative criteria that become visible in interaction and can be grounded more reliably in user behavior.
>
> We believe this focus is well aligned with our motivation. Our goal is not simply to test whether a model completes the requested task, but to capture the gap between apparent task completion and real user utility. Expectations revealed in follow-up interaction are especially valuable for this purpose, because they directly expose where a response that appears complete still falls short of what the user actually values.
>
> **2. Manual Validation of Expectations**
>
> We agree that the quality of extracted expectations is critical, since they serve as the primary reference for benchmark evaluation. We would like to clarify that the current submission already includes multiple safeguards during benchmark construction: (1) data filtering to retain instances suitable for expectation-driven evaluation, (2) explicit extraction constraints that distinguish expectations from restatements, instructions, unsupported inferences, and new requests, and (3) manual review of every instance to verify overall quality and remove noisy or degenerate cases.
>
> To address this concern more directly, we add an extraction-level human validation study on a randomly sampled subset of 100 user queries, comprising 289 individual expectations. Three annotators independently label each expectation, and we use majority voting for the final label. We evaluate two aspects: whether each expectation is supported by the follow-up, and whether it remains anchored to the initial request.
>
> We find that 95.5% are clearly supported by the follow-up, 4.5% are weakly but still plausibly supported, and 0% are unsupported. We further find that the extracted expectations remain strongly grounded in the initial request: 4.5% are Explicit Requirements (directly stated), 73.7% are Implicit Expectations (reasonably inferable), 16.6% are Refinement / Preference (further specification), 5.2% are Context-Utilization Needs (requiring better use of existing context), and 0% are New / Not Anchored. These results directly support the validity of our extraction step.
>
> **3. Metrics in Sec5.1**
>
> We thank the reviewer for raising this important point. We agree that user-specific variation is central in real-world interactions. Our view, however, is that this variation should be understood less as complete randomness and more as the need for finer context understanding and inference. In EXPECTBENCH, the task is not a canonical mapping from a short query to a single fixed expectation set; rather, it is grounded in rich real-world initial queries (366 tokens on average), where different user-valued criteria can arise from subtle contextual differences. We also account for possible remaining stochasticity by making the prediction setting fairly permissive: although each interaction contains only 2.91 expectations on average, we allow models to predict up to 10.
>
> This setup serves two purposes. First, it reflects the real-world question of whether a model can anticipate likely user expectations from the initial query. Second, it helps explain the failure mode behind the results in Section 4. We therefore believe expectation-level prediction remains the more informative primary metric.
>
> We also agree that category-level evaluation is a meaningful complementary view. To further address the reviewer’s concern, we conduct a category-level expectation prediction analysis with GPT-4o on a random sample of 500 instances, mapping both gold and predicted expectations to the discovered taxonomy. The resulting coverage is 42.20% / 55.76% / 67.48% / 73.60% at prediction budgets of 3 / 5 / 8 / 10, respectively. This shows that the same difficulty remains even under the more forgiving category-level metric, and is therefore consistent with our original findings. We further observe an interesting pattern: the model tends to over-predict more surface-level categories, such as Coherence (10.13% vs. 6.39%), suggesting an amplification of surface-level preference bias.
>
> **4. LENS Implementation Details**
>
> We thank the reviewer for pointing this out. Due to space constraints, this concern is addressed in more detail in Reviewer HUtJ, Q5, where we clarify the roles of the Expectation Observer and Expectation Projector, as well as the training procedure. We sincerely thank the reviewer for the understanding.

---

> > ### Author Rebuttal · Reviewer_5asc · 2026-04-04
> >
> > I appreciate the responses, considering all the clarifications and I am maintaining my score.

---

> > > ### Author Response · Authors · 2026-04-06
> > >
> > > We sincerely thank the reviewer again for the thoughtful engagement with our work, including the recognition that the paper identifies a meaningful and underexplored problem and offers a practical methodology. We also appreciate the reviewer taking the time to read our rebuttal.
> > >
> > > We would only like to briefly note a few concrete additions that may be helpful in further clarifying our response. First, some expectations may indeed remain unspoken in later turns because they have already been satisfied, but such expectations are inherently difficult to verify reliably at scale without additional confirmation from the original user. We therefore further clarified why the benchmark focuses on expectations that become visible through interaction and why this setting is especially informative for revealing the gap between apparently completed responses and what users actually value. Second, in addition to the existing filtering, extraction constraints, and manual review, we also added extraction-level human validation to directly verify the quality of the extracted expectations. Third, for Section 5.1, we further clarified the motivation of the setup and added the suggested category-level analysis, which remains consistent with the original finding under a more forgiving metric. Finally, we further clarified the method presentation so that the design and data flow are more explicit.
> > >
> > > We hope these additional clarifications and analyses will be useful in the reviewer’s final consideration of our paper, and we sincerely appreciate the reviewer’s time and care in evaluating our work.

---

### Official Review · Reviewer_HUtJ · 2026-03-16

**Soundness:** 3
**Presentation:** 3
**Significance:** 2
**Originality:** 3
**Overall Recommendation:** 4
**Confidence:** 3

**Summary:**

The authors highlight the issue that expectations are often differ between users and are not explicitly understood or measured for existing systems.

Therefore the authors propose:
1. A benchmark that
i) includes a curated dataset (from a previous dataset, WildChat)
ii) includes an evaluation metric and evaluation method
iii) shows that existing methods do not work well on it.

2. A method that:
i) overcomes existing challenges with expectation aware evaluation

**Compliance With Llm Reviewing Policy:**

Affirmed.

**Final Justification:**

Please see rebuttal acknowledgement.

**Key Questions For Authors:**

Benchmark:
1. What is the cost of running the benchmark? (You do not report that)
2. How does your benchmark compare if evaluating with smaller models? (Not everyone has access to GPT-4o, additionally GPT-4o can be argued as being not scientific as it is a commerical product, rather than a clearly fixed machine learning model).

Method (Lens):
1. There is not enough information provided on what the projector model is exactly doing. What is the input (encoded how) what is the output (encoded how)? The Loss in equation 4) is ambigious and not clear. Overall, even Appendix D does not provide answers to these questions.

**Limitations:**

The authors have a limited limitation discussion. It would be interesting to have an explicit section (maybe in the Appendix) - discussing limitations of the method.

1. E.g. Cost of evaluation
2. What happens when GPT-4o is no longer available, how is evaluation using other models?
3. What are the limitations of the LENS method?

**Strengths And Weaknesses:**

Strengths:
1. Evaluation benchmark introduced seems to be interesting. E.g. it shows an interesting pattern between models (Figure 4).
2. The authors also show in Figure 5 that the benchmark is not fully solved yet.
3. They propose a method that outperforms other baselines on their benchmark, the method also is motivated well by the problem

Weaknesses:
1. It is not quite clear how smaller models / open source models handle the evaluation benchmark (ie. the actual judging of responses). Also, is there a bias if the model is bigger that the responses are better? Also, costs of the benchmark are not reported.
2. The LENS method is evaluated on 7B models only.
3. The LENS method is not described sufficiently.

---

> ### Author Rebuttal · Authors · 2026-03-31
>
> We sincerely thank the reviewer for the insightful feedback and positive evaluation of our work.
>
> **1. Alternative Evaluator**
>
> We agree that evaluator robustness is an important concern. To test this directly, we add Llama-3.3-70B-Instruct as a strong open-weight alternative judge. The resulting agreement remains high: $\kappa = 0.588 / r = 0.94$ against GPT-4o, and $\kappa = 0.543 / r = 0.96$ against human judgments. This shows that our conclusions do not depend on a single proprietary evaluator. More importantly, EXPECTBENCH is evaluator-agnostic by design: the benchmark is grounded in explicit user-derived expectation rubrics, and the judge’s role is only to score responses against these criteria rather than define them. Therefore, changing the evaluator may affect robustness, but not the underlying evaluation target.
>
> **2. Judge Preference for Stronger Responses**
>
> We thank the reviewer for raising this important point. We agree that stronger models may produce better responses overall, as they generally possess stronger underlying capabilities. At the same time, our results suggest that this does not reduce the problem to a small-model artifact. Even the strongest model we evaluate, GPT-4o, achieves an average score of only 2.72 / 5, indicating that most responses still only partially satisfy user expectations. This suggests that the gap we study is not specific to weaker models, but reflects a more systematic expectation-alignment challenge. In other words, larger model scale may improve performance, but it does not eliminate the underlying difficulty of anticipating and satisfying real user-valued criteria.
>
> **3. Benchmark Cost**
>
> We thank the reviewer for raising this practical question. We additionally estimate the cost of running the benchmark. Across the 6,000-instance test split, evaluation requires about 1.79K prompt tokens and 84–95 completion tokens per instance on average, corresponding to an estimated total API cost of approximately \\$32, based on the pricing of \\$2.5 / 1M input tokens and \\$10 / 1M output tokens.
>
> **4. LENS on Stronger Base Model**
>
> We thank the reviewers for raising this important point. We additionally evaluate it with Llama-3.3-70B-Instruct as the main response model and LLaMA-3.1-8B-Instruct as the auxiliary expectation observer. The overall score improves from 2.58 to 2.69, with gains across all ten expectation dimensions: Cx: 2.59→2.71, Cr: 2.45→2.57, Co: 3.06→3.20, Cs: 2.59→2.72, Ct: 2.80→2.95, Ep: 2.82→2.95, Vs: 2.25→2.35, Pr: 2.41→2.52, En: 2.62→2.73, Cm: 2.42→2.53. This suggests that the benefit of LENS does not vanish at larger scale; the latent expectation signal remains useful even for substantially stronger base models.
>
> **5. Method Clarity**
>
> We thank the reviewers for pointing this out. We appreciate the opportunity to clarify the implementation of LENS more concretely. In our framework, the Expectation Observer is a smaller model from the same family as the main response model, and it takes the input query to produce a latent expectation representation. The Expectation Projector is implemented as a lightweight MLP, which maps this observer-side latent representation into a form compatible with the representation space of the main response model, so that it can be used as latent guidance during response generation.
>
> Eq. (4) is the key training objective for the Projector. Concretely, given an input query, the Observer first produces a latent expectation representation, which is then transformed by the Projector and provided to the main response model in the expectation-prediction stage. The main response model predicts the expectation annotations conditioned on both the original query and the projected latent signal, and the negative log-likelihood between these predicted expectations and the gold expectation annotations is used to train the Projector. In this stage, both the Observer and the main response model are kept fixed, and only the lightweight Projector is optimized.
>
> At inference time, the learned Projector maps the observer-side latent signal into the main response model space, and this projected representation is then used as latent guidance for response generation.
>
> We will add these details more explicitly to the main text and Appendix D.

---

> > ### Author Rebuttal · Reviewer_HUtJ · 2026-04-06
> >
> > We thank the authors for the response and additional experiments.
> >
> > Overall, the response has addressed the issue. The method still remains unclear and the paper would benefit from clear definitions of: 1. the entire data flow; 2. each input-output function of each part of the dataflow.
> >
> > We believe our score reflects the overall work, we hope the authors can add the clarificaiton of their method in (even) more detail and clarity to the paper. The work overall represents a sufficient contribution to ICML.

---

> > > ### Author Response · Authors · 2026-04-06
> > >
> > > We sincerely thank the reviewer for the positive follow-up. We are glad that the rebuttal helped address the issues, and we appreciate the opportunity to further clarify the method. Below, we provide clearer definitions of (1) the entire data flow and (2) the input-output function of each component.
> > >
> > > **1. Definition of the Entire Data Flow**
> > >
> > > Given an input query $q$, LENS first feeds $q$ into the **Expectation Observer** $O_{\phi}$, which produces an observer-side latent expectation representation $z(q)$. This representation is then passed to the **Expectation Projector** $P_{\psi}$, a lightweight MLP, which maps $z(q)$ into a projected latent signal $h(q)$ aligned with the representation space of the main response model. In **Stage I**, the main response model is used in an expectation-prediction mode: it takes both the original query $q$ and the projected latent signal $h(q)$ as input, and predicts the corresponding expectation annotations $\tilde{E}(q)$. The negative log-likelihood between $\tilde{E}(q)$ and the gold expectation annotations $E(q)$ is then used to optimize **only the Projector**, while both the Observer and the main response model remain frozen.
> > >
> > > After training, in **Stage II**, for a new query $q'$, the same pipeline is applied again: the Observer first produces $z(q')$, the learned Projector maps it to $h^{\*}(q')$, and the main response model then generates the final response $r(q')$ conditioned on both the query $q'$ and the projected latent signal $h^*(q')$. In this way, the full data flow is:
> > > $$q \rightarrow O_{\phi}(q)=z(q) \rightarrow P_{\psi}(z(q))=h(q) \rightarrow M_{\theta}^{exp}(q,h(q))=\tilde{E}(q)$$
> > > during Stage I, and
> > > $$q' \rightarrow O_{\phi}(q')=z(q') \rightarrow P_{\psi^{\*}}(z(q'))=h^{\*}(q') \rightarrow M_{\theta}^{resp}(q',h^{\*}(q'))=r(q')$$
> > > during Stage II.
> > >
> > > **2. Definition of the Input–Output Function of Each Component**
> > >
> > > - **Expectation Observer** $O_{\phi}$
> > >   **Input:** the user query $q$.
> > >   **Output:** an observer-side latent expectation representation $z(q)$, which captures the expectation-related signal inferred from the query.
> > >
> > > - **Expectation Projector** $P_{\psi}$
> > >   **Input:** the observer-side latent representation $z(q)$.
> > >   **Output:** a projected latent signal $h(q)$ that is compatible with the representation space of the main response model.
> > >
> > > - **Main Response Model in Stage I** ($M_{\theta}^{exp}$)
> > >   **Input:** the original query $q$ together with the projected latent signal $h(q)$.
> > >   **Output:** predicted expectation annotations $\tilde{E}(q)$.
> > >   **Training signal:** the negative log-likelihood between $\tilde{E}(q)$ and the gold expectation annotations $E(q)$ is used to optimize only $P_{\psi}$, while $O_{\phi}$ and $M_{\theta}$ are fixed.
> > >
> > > - **Main Response Model in Stage II** ($M_{\theta}^{resp}$)
> > >   **Input:** a new user query $q'$ together with the projected latent signal $h^*(q')$.
> > >   **Output:** the final generated response $r(q')$.
> > >
> > > We hope this clarifies the full data flow and the role of each module more concretely. We will also incorporate this level of explicit definition into the revised paper.

---

### Decision · Program_Chairs · 2026-04-30

**Decision:**

Accept (regular)

**Comment:**

This paper proposes EXPECTBENCH, a benchmark grounded in real user expectations extracted from multi-turn LLM conversations, and LENS, a lightweight latent expectation-aware response generation framework. The benchmark reveals a systematic gap: even GPT-4o achieves only 2.72/5 on expectation satisfaction. 3 of 4 reviewers recommend acceptance， as the paper provides a valuable benchmark resource and a well-motivated method for the alignment community.